# Development of mRNA Vaccines: Scientific and Regulatory Issues

**DOI:** 10.3390/vaccines9020081

**Published:** 2021-01-23

**Authors:** Ivana Knezevic, Margaret A. Liu, Keith Peden, Tiequn Zhou, Hye-Na Kang

**Affiliations:** 1Department of Health Product Policy and Standards, World Health Organization, Avenue Appia 20, CH-1211 Geneva, Switzerland; zhout@who.int (T.Z.); kangh@who.int (H.-N.K.); 2ProTherImmune 3656 Happy Valley Road, Lafayette, CA 94549, USA; 3Office of Vaccines Research and Review, Center for Biologics Evaluation and Research, US Food & Drug Administration, Silver Spring, MD 20993, USA; keith.peden@fda.hhs.gov

**Keywords:** WHO standards, mRNA vaccines, prophylactic vaccines, vaccine development, regulatory considerations

## Abstract

The global research and development of mRNA vaccines have been prodigious over the past decade, and the work in this field has been stimulated by the urgent need for rapid development of vaccines in response to an emergent disease such as the current COVID-19 pandemic. Nevertheless, there remain gaps in our understanding of the mechanism of action of mRNA vaccines, as well as their long-term performance in areas such as safety and efficacy. This paper reviews the technologies and processes used for developing mRNA prophylactic vaccines, the current status of vaccine development, and discusses the immune responses induced by mRNA vaccines. It also discusses important issues with regard to the evaluation of mRNA vaccines from regulatory perspectives. Setting global norms and standards for biologicals including vaccines to assure their quality, safety and efficacy has been a WHO mandate and a core function for more than 70 years. New initiatives are ongoing at WHO to arrive at a broad consensus to formulate international guidance on the manufacture and quality control, as well as nonclinical and clinical evaluation of mRNA vaccines, which is deemed necessary to facilitate international convergence of manufacturing and regulatory practices and provide support to National Regulatory Authorities in WHO member states.

## 1. Introduction

A number of mRNA candidates have entered clinical trials for applications as diverse as prophylaxis of viral diseases and immunotherapies for various types of cancer [1,2]. Emerging infectious diseases present compelling targets for the nascent technology for a variety of reasons, including the rapidity with which they can be made when faced with a newly discovered pathogen. For this reason, targets with candidates in clinical trials have included influenza viruses (H7N9 and H10N8) and Zika virus, and most recently, and most urgently, the causative agent for COVID-19, SARS-CoV-2. The prophylactic vaccine candidates use mRNA formulated in lipid nanoparticles (LNPs). Table 1 provides a listing of clinical trials that have been undertaken for prevention of infectious diseases with mRNA prophylactic vaccines. The onset of the pandemic caused by SARS-CoV-2 resulted in massive efforts to develop COVID-19 vaccines via a number of different technologies; due to the rapidity with which mRNA vaccine candidates could be designed and manufactured, they were among the first COVID-19 vaccines to enter clinical trials. Indeed, the most advanced mRNA vaccine candidates are COVID-19 vaccines, which include Phase 3 clinical trials (see Table 2). 

The WHO Research and Development (R&D) Blueprint is a global strategy and preparedness plan that allows for the rapid activation of R&D activities. For the purposes of the R&D Blueprint, WHO has developed a special tool for determining which diseases and pathogens to prioritize for research and development in public health emergency (PHE) contexts [3,4]. At present, the priority diseases are: COVID-19, Crimean-Congo haemorrhagic fever, Ebola virus disease and Marburg virus disease, Lassa fever, Middle East respiratory syndrome coronavirus (MERS-CoV) and Severe Acute Respiratory Syndrome (SARS), Nipah and henipaviral diseases, Rift Valley fever, Zika, “Disease X”. “Disease X” represents the knowledge that a serious international epidemic could be caused by a pathogen currently unknown to cause human disease. The R&D Blueprint explicitly seeks to enable early cross-cutting R&D preparedness that is also relevant for an unknown “Disease X”. This is not an exhaustive list, nor does it indicate the most likely causes of the next epidemic. WHO reviews and updates this list as needs arise and methodologies change. Based on the priority diseases, WHO then works to develop R&D roadmaps for each one.

The WHO Director General declared that the outbreak of COVID-19 constitutes a Public Health Emergency of International Concern in January 2020 and assessed that COVID-19 can be characterized as a pandemic in March 2020. mRNA vaccines were among the first candidates to enter clinical development during the COVID-19 pandemic and are expected to become the first licensed vaccines. Two mRNA vaccines have demonstrated around 95% efficacy in Phase 3 clinical trials [5,6] and have recently been approved for emergency use by national regulatory authorities in some countries to combat the pandemic [7,8]. On 31 December 2020, the WHO listed a COVID-19 mRNA vaccine for emergency use, making the Pfizer/BioNTech vaccine the first to receive emergency validation from WHO [9]. It is expected that mRNA vaccines will become a platform technology that offers the potential to develop vaccines quickly against infection with priority pathogens in a PHE. This article describes the current status of mRNA vaccines (messenger RNA (mRNA) vaccines). The role WHO has in improving regulatory convergence at the global level by developing standards is also discussed, since promoting regulatory convergence is recognized as a key enabler in the R&D Blueprint. 

## 2. Development of mRNA Vaccines

The demonstration of the use of mRNA as a potential in-vivo gene-delivery technology was published in 1990, when the direct injection of “naked” mRNA was shown to be capable of resulting in in vivo expression of the encoded protein [10]. However, various issues hindered the immediate ability to use in vitro transcribed mRNA as a facile means to generate protein immunogens in vivo following simple injection. These included the instability of mRNA in vivo, due to the near-ubiquitous presence of RNases. In addition, the mRNA was quite immunogenic, stimulating innate responses with a concomitant decrease in translation of the mRNA. While innate immune responses might be beneficial for vaccine applications of mRNA, the stability and hence mRNA production challenges still remained. 

A significant advance that led to the rapid expansion of efforts to use mRNA as a platform technology was the discovery by Karikó and Weissman that the use of modified nucleosides resulted in a decrease in the immunostimulatory effects of the in-vitro transcribed mRNA, via a decrease in Toll-like receptor (TLR) stimulation (as further explained below in Section 3) [11]. They went on to show that, by the use of pseudouridine in place of uridine, the resulting mRNA was also more stable and had increased translational capability [12]. A further development by Schlake and colleagues [13] enabled the production of mRNA that similarly did not stimulate the innate responses and avoided decreased protein production simply with sequence engineering, without the need for nucleoside modification. The development of formulations (mainly LNPs) that both help stabilize the mRNA and facilitate its delivery into cells and release from endosomes [14] (and likely act as adjuvants for the encoded protein [15]) has also been crucial. LNPs are composed of various lipids, often including phospholipids, cholesterol, ionic lipids, and polyethylene glycol-conjugated lipid, which form to have an aqueous center in which the charged mRNA molecules are located. The mRNA is thus protected, and the lipid particle facilitates entry into cells and even exit from lysosomes for delivery of the mRNA, as further described by Reichmuth and colleagues [14]. Manufacturers have their own proprietary LNPs with differing characteristics and methods of manufacture. Because new LNP formulations will be developed (such as one manufacturer is currently doing to improve thermostability), and because the actual composition (such as percentage of each component) and the manufacturing process is proprietary, this information is beyond the scope of this manuscript. 

Note that while the focus of most vaccine candidates that are using mRNA technology has been on antibody responses, T-cell responses have also been shown to be generated by mRNA vaccines. These include CD4^+^ T-helper cells, which are needed for optimal antibody responses [16,17]. The role of T cells in the assessment of vaccine efficacy may be difficult to determine (in terms of any correlate of immunity), as no vaccines based solely on T-cell responses have been developed. Nevertheless, the role of both types of T cells in both generating antibody responses (T-cell help) and their ability to kill virally-infected cells (cytolytic T lymphocytes) add this additional arm of immunity to mRNA vaccines. For example, anti-SARS-CoV-2 T cells may be important, since neutralizing antibodies wane over time [18,19]. 

Another type of mRNA vaccine is called self-amplifying RNA vaccine, earlier versions of which have been made and studied pre-clinically with plasmid DNA vaccines and with viral vectors (reviewed in [1]). A self-amplifying mRNA (samRNA) vaccine encodes both the desired antigen and key viral replicon proteins derived from a different virus, such as alphaviruses, and not from the target virus. The production of the alphaviral replicon encoded by the mRNA results in the transduced cell then being able to produce many copies of the antigen mRNA, and hence much more of the protein antigen than would usually be produced per molecule of mRNA encoding the antigen. This is done without making a whole virus, since the mRNA only codes for the antigen and the specific replicon proteins. Because many more copies of the protein antigen will be made, such a self-amplifying mRNA vaccine may be more potent and might not require a booster dose.

A significant reason for the enthusiastic embrace of mRNA for a COVID-19 vaccine is the speed with which a vaccine candidate can be generated, since the mRNA construct can be generated based on knowing a pathogen’s genetic sequence and the antigen to target. Another reason for the interest is that the manufacturing process is one that is essentially generic for mRNA vaccines and independent of the antigen encoded by the vaccine. 

### 2.1. Manufacturing of mRNA Vaccines

The manufacturing process involves in-vitro transcription of a DNA template that contains the mRNA sequence. This mRNA has standard elements, such as a 5′ untranslated region (UTR), a start codon and an open reading frame (ORF), a stop codon followed by a 3′ UTR and a poly(A) sequence (see Figure 1). The ORF is frequently optimized for expression (both codon optimization, and optimization of RNA for translation and stability have been used). The mRNA is capped and polyadenylated to allow efficient translation. The cap can be added during transcription or post transcriptionally; the poly(A) sequence can be added during transcription or be added post transcriptionally with poly(A) polymerase. The sequences of the 5′ and 3′ UTRs can have significant effect on translation efficiency. 

The process starts with a linearized DNA template, and a DNA-dependent RNA polymerase (usually from phage T7) transcribes the template into mRNA using nucleoside triphosphates as the building blocks. The nucleosides may be native nucleosides or naturally-occurring modified nucleosides [1,2,12,13]. It is important during mRNA purification to remove the DNA template, enzymes, and nucleotides. Different manufacturers have optimized their particular constructs in terms of the nucleosides used (i.e., how modified, if they are modified), the sequence, and for the lipid nanoparticle formulations to achieve the appropriate immunogenicity, potency, and thermostability has been recently summarized [20]. In addition, both in pre-clinical and clinical studies, the goal is finding the optimal level of immune stimulation while limiting any clinical adverse events (such as injection-site soreness, or systemic symptoms like chills) due to the strong immune responses. 

### 2.2. Immunological Potential of mRNA Vaccines

mRNA stimulates innate-immune responses, and a variety of cellular pathways are activated, including Toll-Like Receptors: TLR3, TLR7, and TLR8 of the innate immune system, as well as via various cytoplasmic proteins, notably PKR (Protein Kinase R), RIG-I (Retinoic Acid-Inducible Gene I), OAS (2′-5′-Oligoadenylate synthetases), and MDA5 (Melanoma Differentiation-Associated protein 5) via cytoplasmic proteins [21]. While the use of modified nucleosides has resulted in a decrease in the immunostimulatory activity of mRNA, inflammation resulted in some non-serious, adverse events clinically, which are described in the reports of the clinical trials [16,17], and resulted in selecting the second highest dose in two cases [16,17]. The vaccine industry, scientific community and public health officials have discussed the possibility of lowering the dose of COVID-19 mRNA vaccine compared with the dosage originally evaluated in Phase 3 trials. This would significantly increase the number of available doses providing that it does not affect vaccine efficacy.

Of note is the fact that self-amplifying RNA vaccines generally are not produced with modified nucleosides, since the RNA amplified in the cell will be synthesized with unmodified nucleosides. 

The rapidity with which clinical trials are progressing for COVID-19 vaccines and for potential worldwide usage have created a need for the WHO to address the relevant characteristics of mRNA vaccines in order to evaluate the quality, safety, and efficacy of mRNA for prophylactic vaccines. The inherent immunological and structural properties of mRNA, the need for formulation for delivery, and the manufacturing process all need to be considered, even though certain information for any specific vaccine candidate may be proprietary and thus not generally known at this time in order to set specific guidance. In addition to the more general issues that need to be considered for prophylactic vaccines, there are certain considerations that may be specific for COVID-19 vaccines, due to the specific and protean pathophysiologic manifestations that have been seen in severe infection with SARS-CoV-2. One of these considerations is the potential for vaccine-enhanced respiratory disease (VERD), which has been seen with some vaccines for other viruses [22] and appears to be minimised with vaccines that induce at Th1-type immune response. 

## 3. Regulatory Considerations

Many of the regulatory issues required for mRNA vaccines are similar to those for any vaccine such as quality of the starting materials, consistency of manufacture, demonstrated evidence of safety and efficacy during pre-clinical studies, clinical trials and post-marketing surveillance. Some recent reviews have discussed regulatory issues for the manufacture, quality control, and safety for mRNA vaccines [23,24,25]. Nevertheless, because the production of mRNA vaccines involves new technologies, some of the regulatory considerations are specific to this class of vaccine. While the process of producing the mRNA (or samRNA) seems to be coalescing around enzymatic synthesis of the RNA from a linear DNA template using a phage DNA-dependent RNA polymerase and complexing the RNA in lipid nanoparticles, the process details generally remain proprietary and not available to the public. Thus, at this time, it is not possible to define specific principles for evaluation of mRNA vaccines. Therefore, regulatory requirements should be established by national regulatory authorities on the basis of WHO Guidelines or Recommendations for evaluation of vaccines. The quality and consistency of the enzymes, nucleotides, and the linear DNA templates are critical. The raw materials such as the lipids should be carefully controlled and their purity demonstrated. The linear DNA template should be generated under current Good Manufacturing Practice (cGMP), although at early stages of vaccine development, this may not be necessary. Some of the important considerations for the Drug Substance are the purity of the RNA, with the proportion of full-length transcripts and the amounts of short transcripts and double-stranded RNA present being critical parameters to monitor, as is the percentage of mRNA that is capped and polyadenylated. For the Drug Product, however it is produced, it is important to quantify the amount of mRNA encapsulated in the particle, and determine the size distribution of the particles, which is used to monitor the consistency of manufacture. Some of the usual parameters to monitor include stability, identity, and sterility. Thermal stability of mRNA vaccines is one of the challenges for vaccine developers, regulators and users. A cold-chain requirement is considered as one of the limitations for the use of mRNA vaccines particularly in low- and middle-income countries, where it might be difficult to ensure transportation and storage of vaccines under a cold chain. Assays to determine potency for this platform are under development and should be discussed with the national regulatory authority, but cell-free translation of mRNA extracted from the Drug Product and transfection of cells in culture with the Drug Product and detection of the antigen by some immunological method are two commonly used methods to ascertain that the mRNA encodes the desired antigen. 

In the context of animal studies and clinical trials in humans, monitoring of systemic and local toxicity and inflammatory responses are expected. If needed, dose lowering may need to be considered. Apart from demonstrated issues with mRNA vaccines, there are also some perceptions that regulators need to address. New production technologies usually raise some questions about the safety of the novel products. One of the concerns expressed is related to the potential risk of the RNA integrating into human DNA. It should be noted that while there was a theoretical concern for integration into the host genome with regard to plasmid DNA vaccines, this concern is not shared for mRNA-based vaccines for the following reasons. (1) mRNA remains in the cytoplasm and is not transported to the nucleus. (2) For integration, the mRNA likely needs to be converted to a DNA molecule. Although it is possible that single-stranded DNA can be integrated, the integrating form of DNA is generally double-stranded DNA. This requires the presence of a reverse transcriptase and appropriate primers and complementary binding sites on the mRNA to generate first a single-stranded DNA and then convert this single-stranded DNA to a double-stranded DNA, which again requires appropriate primers and binding sites. In retroviruses, this process occurs in the retrovirus particle. This is not the case with mRNA vaccines, for while endogenous reverse transcriptases are present in mammalian cells, the enzymes and RNA are not found in the appropriate complex to allow efficient reverse transcription to occur. (3) The final step in retroviral integration requires the activity of the viral integrase, which again is in the retroviral particle. Integration of naked double-stranded DNA has been shown to be very inefficient. (4) The vaccine mRNA has been shown to degrade within a relative short time once taken up in the body’s cells. Finally, because cellular mRNA is more abundant than a mRNA vaccine, it is highly unlikely that a cellular reverse transcriptase (RT) would preferentially copy the vaccine mRNA over cellular mRNA. For all these reasons, the integration risk of mRNA vaccines has been considered negligible. 

## 4. WHO Standards for Regulatory Evaluation of Vaccines and Other Biologicals

As part of its mandate, WHO has a unique role to support regulatory authorities in its 194 member states. One of WHO core functions is “setting norms and standards and promoting and monitoring their implementation”. The Expert Committee on Biological Standardization (ECBS) has been active in establishing WHO standards for biologicals for more than 70 years. 

WHO standards, both written and measurement, are based on scientific evidence and provide the basis for establishing and updating national regulatory requirements. They also provide a global perspective and are published in the WHO Technical Report Series. The role of the international recommendations or guidelines for biological substances is to ensure the availability of vaccines of assured quality, safety and efficacy for use in international immunization programmes. Furthermore, these documents serve as a benchmark for global acceptability of products and as a basis for defining national regulatory requirements for licensing as well as for post-licensure evaluation. These activities include both measurement (physical standards) and written standards for vaccines, biotherapeutic products and other biologicals, intended as guidance for National Regulatory Authorities and manufacturers. The development of measurement standards involves elaborate collaborative studies in numerous laboratories worldwide, and the WHO written standards are based on scientific consensus achieved through much international consultation. The work is supported by WHO Collaborating Centres, National Regulatory Authorities in many countries, pharmacopoeias, manufacturers associations, and academia. 

As examples of the measurement standards for COVID-19, the First WHO International Standard of anti-SARS-CoV-2 immunoglobulin with assigned unitage of 250 IU/ampoule (neutralizing antibody activity) and the First WHO International Reference Panel of anti-SARS-CoV-2 immunoglobulin were established by WHO ECBS on 10th December 2020. These standards are intended as global reference reagents against which national reference preparations would be calibrated. Calibration of national references against a single global standard will facilitate comparison of results of assays (e.g., of the antibody response to candidate COVID-19 vaccines) conducted in different countries. The development and scientific assessment through collaborative studies of these reagents have been completed in record time, and the standards were made available by WHO Collaborating Centre, National Institute for Biological Standardization and Control (NIBSC), the United Kingdom, at the end of December 2020. It is expected that the use of these standards would contribute to the better understanding of the immune response, particularly correlates of protection, which are of great importance for the success of immunization. 

WHO also plays an important role in the implementation of new guidelines and recommendations that also serve as tools for global regulatory convergence. Detailed information about WHO international standards for biologicals can be found at the WHO biologicals website [26].

In addition, WHO has other initiatives that are closely linked to the standardization of vaccines. In particular, strengthening of national regulatory authorities is one of the important elements in assuring the quality of vaccines worldwide. Prequalification of vaccines is an important mechanism through which vaccines prequalified by WHO become subject of the United Nations Children’s Fund (UNICEF) supply. The safety of vaccines and the issues discussed at the WHO Global Advisory Committee on Vaccine Safety are also important as well as WHO activities related to the immunization policy led by the Strategic Advisory Group of Experts. In the context of all these activities, mRNA vaccines need to be considered as new immunization tools of critical importance for the control of the pandemic caused by SARS-CoV-2 with an expectation that vaccines produced with this technology will also play a role in the control of other diseases. 

## 5. Way Forward

In view of the development of mRNA vaccines and scientific and regulatory challenges faced, it is considered necessary to establish an international consensus on the technical expectations for the development and evaluation of mRNA vaccines. This will facilitate international convergence of manufacturing and regulatory practices for mRNA vaccines worldwide, and provide technical assistance to countries. The availability of WHO standards will also facilitate WHO prequalification or Emergency Use Listing assessment of such vaccines, in particular in the case of current COVID-19 pandemic. 

Although there are a number of guidance documents available from WHO in terms of assuring the quality, safety and efficacy of vaccines, there are currently no guidelines specifically for vaccines based on RNA platforms. The WHO ECBS discussed these issues at its meeting in August and December 2020 and supported the development of a document on regulatory considerations for the evaluation of mRNA vaccines, which could be updated as more scientific and clinical data became available. 

In urgent response to the current COVID-19 pandemic, as of end December 2020, some mRNA vaccines have been the subject of review and regulatory approval for emergency use. However, ongoing and future studies are needed to address unknowns and gaps of evidence that are critical for completing the data for the purpose of licensing these vaccines. The step from the approval for emergency use to the licensing may require a number of additional studies depending on the intended use and supportive data for demonstration of the quality, safety and efficacy of vaccines.

WHO endeavors to convene international discussions among experts of vaccine developers, manufacturers and regulators to review available scientific evidence, discuss key issues and reach consensus on scientific and technical expectations towards assuring the quality, safety and efficacy of mRNA vaccines. At the time of writing this paper, WHO assembled a drafting group composed of experts from the scientific field of mRNA vaccines and regulatory authorities to develop a draft document that will cover manufacturing and quality, non-clinical and clinical issues of mRNA vaccines. In addition, a number of regulatory issues and challenges would be elaborated in that document to provide information about current regulatory thinking on the key principles for evaluation of these vaccines. WHO will convene public consultations as well as consultations with stakeholders starting from January 2021 to seek critical review of the draft document and invite comments. The final document, formulated on the basis of scientific evidence and broad consensus, taking into consideration input from stakeholders, will be submitted to the ECBS for review in late 2021.

Fundamentally important, vaccine developers, especially those that are at advanced levels of development, e.g., entered Phase 2 and/or Phase 3 clinical trials and thus may have proof of vaccine safety and efficacy, are strongly encouraged to share with the public the data for candidate mRNA vaccines in terms of quality, safety and clinical aspects. Various global or regional regulatory forums are already available to discuss issues relevant to the quality, safety and efficacy of the products across nations/agencies. Meanwhile, technologies are also advancing (e.g., optimization of production, new technology) along with better understanding of the target pathogens and the mRNA platforms. These are all important factors to be taken into consideration when formulating WHO guidance for mRNA vaccines.

It is hoped that in the near future more data on mRNA vaccines in terms of their quality, safety and efficacy will be available to the public, and an international consensus will be reached among stakeholders on the technical specifications of mRNA vaccines towards assured quality, safety and efficacy. It is expected that safe and efficacious mRNA vaccines will play an important role in combating infectious diseases in response to public-health emergencies. Building trust in mRNA vaccines at the global level requires careful review of scientific issues specific for this type of vaccine. This initiative will encompass regular scientific updates in line with the increasing knowledge and understanding of the vaccine performance in the prophylaxis and control of the diseases. WHO standards for regulatory evaluation of mRNA vaccines would provide the basis for establishing or updating national regulatory requirements for evaluation of these vaccines. Access to vaccines of assured quality, safety and efficacy is one of WHO goals, which applies to mRNA vaccines, as well as to other types of vaccines.

## Figures and Tables

**Figure 1 vaccines-09-00081-f001:**
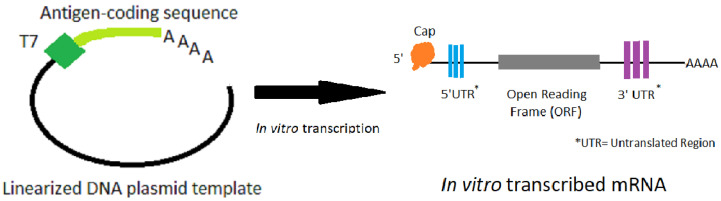
The process of in vitro transcription of mRNA made from a DNA plasmid template that has been linearized.

**Table 1 vaccines-09-00081-t001:** mRNA prophylactic vaccines for infectious diseases in clinical trials.

Product Company/ Institution	Indication (Disease)	National Clinical Trial Identifier	Active Substance: mRNA	Antigen	Formulation	Phase
**RNActive^®^** CureVac	Rabies	NCT02241135 EudraCT 2013-002171-17	CV7201	Rabies virus G protein	Lipid Nanoparticles	1
**mRNA-1440** Moderna (Valneva)	Influenza H10N8	NCT03076385 EudraCT 2015-003452-48	Also referred to as: VAL-506440	Influenza Hemagglutinin H10N8 (A/Jiangxi-Donghu/346/2013)	Lipid Nanoparticles	1
**mRNA-1325** Moderna/BARDA	Zika	NCT03014089	mRNA-1325	prM and E	Lipid Nanoparticles	1
**mRNA-1388** Moderna (Valneva)/ DARPA	Chikungunya	NCT03325075	Also referred to as: VAL-181388	“Viral antigenic proteins” Program appears to have been replaced by mRNA encoding a monoclonal antibody	Lipid Nanoparticles	1
**mRNA-1653** Moderna	Human Metapneumovirus and Human Parainfluenza Virus 3	NCT03392389	mRNA-1653	F protein of each virus	Lipid Nanoparticles	1
**mRNA-1851** Moderna	Influenza H7N9	NCT03345043	Also referred to as: VAL-339851	(Influenza Hemagglutinin H7N9 A/Anhui/1/2013	Lipid Nanoparticles	1
**mRNA-1647** and **mRNA-1443** Moderna	Cytomegalovirus	NCT03382405	mRNA-1647 and mRNA-1443	mRNA-1647: gB (1 mRNA), pentameric complex (5 mRNAs), and mRNA-1443 is pp65	Lipid Nanoparticles	1
**CV7202** CureVac	Rabies	NCT03713086EudraCT #: 2017-002856-10	CV7202	RABV-G	Lipid Nanoparticles	1
**mRNA-1893** Moderna/ BARDA	Zika	NCT04064905	mRNA-1893	preM.E with a JEV leader sequence	Lipid Nanoparticles V1GL	1
**mRNA-1647** Moderna	CMV (tested in seronegative and seropositive patients)	NCT04232280	mRNA-1647	mRNA-1647: gB (1 mRNA), pentameric complex (5 mRNAs)	Lipid Nanoparticles V1GL	2
**GSK3903133A** GSK	Rabies	NCT04062669	Rabies G SAM (CNE)	Rabies G protein (self-amplifying construct)	Lipid Nanoparticles	1
**mRNA-1653** Moderna	Human Metapneumovirus and Human Parainfluenza Virus 3	NCT04144348	mRNA-1653	F proteins of each virus	Lipid Nanoparticles V1GL	1b
**mRNA-1345** Moderna	RSV in older adults	NCT04528719	mRNA-1345	Presumed to be F protein based on pre-clinical publications for other earlier Moderna RSV vaccine programs	Lipid Nanoparticles	1

**Table 2 vaccines-09-00081-t002:** Clinical development of mRNA vaccine candidates against Covid-19.

Product Company/ Institution	National Clinical Trail Identifier	Active Substance: mRNA	Antigen	Formulation	Phase
**BNT162a1, BNT162b1, BNT162b2, BNT162c2** BioNTech/ Pfizer	NCT04380701 EUCT: 2020-001038-36 WHO UTN: U1111-1249-4220 ChiCTR: 2000034825	BNT162a1, BNT162b1, BNT162b2, BNT162c2; uridine-containing, nucleoside-modified, BNT162c2 is self-amplifying	162a1 and BNT162b1 encode trimerized spike protein RBD; 162b2 and 162c2 encode P2-mutated full spike protein; 162a1: unmodified nucleosides, 162b1 and 162b2: modified nucleosides: 1-methyl pseudouridine; 162c2: self-amplifying mRNA (based on VEEV)	Lipid Nanoparticle A30-L-A70	1/2
**BNT162b1** BioNTech/ Pfizer	NCT04523571	BNT162b1 see above	encodes trimerized spike protein RBD	Lipid Nanoparticles	1
**BNT162b2** BioNTech/ Pfizer	NCT04588480	BNT162b2	encodes P2-mutated full spike protein	Lipid Nanoparticles	1/2
**BNT162b3** BioNTech/ Pfizer	NCT04537949 EUCTR2020-003267-26-DE	BNT162b3 nucleoside-modified mRNA	Unknown	Lipid Nanoparticles	1/2
**BNT162b2** BioNTech/ Pfizer	NCT04649021	BNT162b2 see above	encodes P2-mutated full spike protein	Lipid Nanoparticles	2
**BNT162b1, BNT162b2** BioNTech/ Pfizer	NCT04368728 EUCT 2020 = 002641-42	BNT162b1 and BNT1b2b2 see above	see above	Lipid Nanoparticles	2/3
**mRNA-1273** NIAID (Moderna)	NCT04283461	mRNA-1273	Full-length, prefusion-stabilized spike protein	Lipid Nanoparticles	1
**mRNA-1273** Moderna/ BARDA	NCT04405076	mRNA-1273	Full-length, prefusion-stabilized spike protein	Lipid Nanoparticles	2
**mRNA-1273** Moderna/ BARDA/ NIAID	NCT04470427	mRNA-1273	Full-length, prefusion-stabilized spike protein	Lipid Nanoparticles	3
**mRNA1273** Moderna/ BARDA	NCT04649151	mRNA-1273	Full-length, prefusion-stabilized spike protein	Lipid Nanoparticles	2/3
**CVnCoV** CureVac/ CEPI	NCT04449276EUCT 2020-001286-36	CVnCoV	Spike protein	Lipid Nanoparticles	1
**CVnCoV** CureVac	NCT04515147	CVnCoV	Spike protein	Lipid Nanoparticles	2
**CVnCoV** CureVac	NCT04652102 EUCT 2020-001646-20	CVnCoV	Spike protein	Lipid Nanoparticles	2b/3
**CVnCoV** CureVac	EU2020-004066-19	CVnCoV	Spike protein	Lipid Nanoparticles	3
**ARCT-021** Arcturus/ Duke/NUS	NCT04480957	Self-replicating (replicon) mRNA ARCT-021	Prefusion Spike Protein	Lipid Nanoparticle	1/2
**ARCT-021** Arcturus/ Duke/NUS	NCT04668339	Self-replicating (replicon) mRNA ARCT-021	Prefusion Spike Protein	Lipid Nanoparticle	2
**LNP-nCoVsaRNA** Imperial College London	ISRCTN17072692 [EUCT 2020-001646-20]	nCoVsaRNA: self-amplifying mRNA encoding spike protein	Spike protein	Lipid Nanoparticles	1
**ArCoV** PLA Acad Mil Sci/Walvax Biotech Abogen	ChiCTR2000034112	ArCoV	Spike protein	Not listed	1
**ArCoV** PLA Acad Mil Sci/Walvax Biotech AbogenCoV	ChiCTR2000039212	ArCoV	Spike protein	Not listed	1b

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
