# Peer review of "Development of mRNA Vaccines: Scientific and Regulatory Issues"

_vaccines, 2021, doi:10.3390/vaccines9020081_

Round 1

Reviewer 1 Report

The paper from Knezevic et al present a WHO perspective on the scientific and regulatory issues associated with the development of mRNA vaccines, which of course are currently in the news for COVID-19 prophylaxis. Overall, this is a well-written paper, if a little bogged down at times with regulatory details that the general reader might not find interesting. I have some suggestions for improving the article, with the general reader in mind, as follows:

  1. The article mentions lipid nanoparticles a lot, but does not describe exactly what they are; it might be useful to add some description of LNPs to the section on manufacture.
  2. Tables 1 and 2 would benefit from knowing the country of origin of these mRNA vaccines.
  3. Page 5: I would suggest strongly having a Table 3 that lists the disease and pathogens that the WHO thinks are priority.
  4. Second paragraph, page 5: in Dec2020, WHO listed a covid-19 mRNA vaccine for emergency use? Authors clarify and name the vaccine (Pfizers?)
  5. Development of mRNA vaccines section - page 5 to 7.
    a) NLPs more details please. What lipids and how are they particulate in nature?
    b) Sentence added about antibody decline would be useful.
    c) nature of antibody response to mRNA vaccines: afyter 1 dose, is it just IgM and switch to IgG after 2 doses, or does 1 dose generate sufficient IgG?
  6. I thin k a subsection heading on Manufacturing would benefit the reader in breaking up this text. Also the manufacturing section would benefit from using a diagram or flow-chart to describe the process - it's much more informative than dense text.
  7. Page 7: define the acronyms PKR, RIG-1, OAS, mDA5, etc...
  8. Page 7 fist paragraph: authors write of 'discussions are now ongoing'. Between whom - please clarify. Also the issue about the half-dose: is this drawing a conclusion from the clinical trials from the UK with the adenovirus vaccines suggesting full doses may be unnecessary (not so)?
  9. Spell mistake - DNA temples, you mean templates (page 7)
  10. Page 9; reference to NIBSC - I presume you mean NIBSC, UK? 
  11. Way forward section - there is a lot of repetition in this section of previous material and should be removed and the section shortened. 

Author Response

Dear Editor,

Thank you and to the reviewers for taking the time in reviewing our manuscript. Please find below our point-by-point replies to the reviewer’s very helpful comments.

Reviewer #1 (Comments and Suggestions for Authors):

The paper from Knezevic et al present a WHO perspective on the scientific and regulatory issues associated with the development of mRNA vaccines, which of course are currently in the news for COVID-19 prophylaxis. Overall, this is a well-written paper, if a little bogged down at times with regulatory details that the general reader might not find interesting. I have some suggestions for improving the article, with the general reader in mind, as follows:

Author response: We greatly appreciate the reviewer’s suggestions to make the article more accessible for the general reader, and have tried to do so based on the very helpful suggestion. By way of explanation, though, we wanted to point out that since this article is part of a special issue on mRNA vaccines, that we were specifically focused on the perspective that WHO could provide as the international organization that is drafting guidelines. Nevertheless, the reviewer makes an excellent point, and we have tried to modify the article to be more appropriate for general readers.

  1. The article mentions lipid nanoparticles a lot, but does not describe exactly what they are; it might be useful to add some description of LNPs to the section on manufacture.

Author response: The authors agree with the reviewer that additional information regarding the lipid nanoparticles would be useful. The following description was added to the page 6 of the updated version of the manuscript:

“LNPs are composed of various lipids, often including phospholipids, cholesterol, ionic lipids, and polyethylene glycol-conjugated lipid, which form to have an aqueous center in which the charged mRNA molecules are located. The mRNA is thus protected and the lipid particle itself facilitates entry into cells and even exit from lysosomes for delivery of the mRNA, as further described by Reichmuth and colleagues (14). Manufacturers have their own proprietary LNPs with differing characteristics and methods of manufacture. Because new LNP formulations will be developed (such as one manufacturer is currently doing to improve thermostability), and because the actual composition (such as specific lipids and percentage of each component) and the manufacturing process is proprietary, more detail for specific LNPs is beyond the scope of this manuscript.”

  1. Tables 1 and 2 would benefit from knowing the country of origin of these mRNA vaccines.

Author response: The country of origin is not listed due to the fact that the limited number of mRNA vaccines in advanced development are from a restricted number of companies whose origins are well known: Germany, USA, UK, Singapore, China. Furthermore, the inclusion of the country of origin in the tables might be misleading since at least one is made by two companies working together, and different manufacturing sites might be utilized even for the same vaccine.

  1. Page 5: I would suggest strongly having a Table 3 that lists the disease and pathogens that the WHO thinks are priority.

Author response: The authors agree with the suggestion to add WHO priority pathogens and diseases in the textual form rather than another table. The list is useful, but we were concerned that because these priority pathogens change in real time, a table used as an entity might be out of date at the time of reading. Please note that the references do supply the links to the WHO webpages, which will be the up-to-date. The following text is included on the page 5 of the updated manuscript:

“At present, the priority diseases are: COVID-19, Crimean-Congo haemorrhagic fever, Ebola virus disease and Marburg virus disease, Lassa fever, Middle East respiratory syndrome coronavirus (MERS-CoV) and Severe Acute Respiratory Syndrome (SARS), Nipah and henipaviral diseases, Rift Valley fever, Zika, “Disease X”. “Disease X” represents the knowledge that a serious international epidemic could be caused by a pathogen currently unknown to cause human disease. The R&D Blueprint explicitly seeks to enable early cross-cutting R&D preparedness that is also relevant for an unknown “Disease X”. This is not an exhaustive list, nor does it indicate the most likely causes of the next epidemic. WHO reviews and updates this list as needs arise, and methodologies change. Based on the priority diseases, WHO then works to develop R&D roadmaps for each one.”

  1. Second paragraph, page 5: in Dec2020, WHO listed a covid-19 mRNA vaccine for emergency use? Authors clarify and name the vaccine (Pfizers?)

Author response: The name of the company was added on page 5 of the updated manuscript.

  1. Development of mRNA vaccines section - page 5 to 7.
  2. a) NLPs more details please. What lipids and how are they particulate in nature?
    b) Sentence added about antibody decline would be useful.
    c) nature of antibody response to mRNA vaccines: afyter 1 dose, is it just IgM and switch to IgG after 2 doses, or does 1 dose generate sufficient IgG?

Author response:

5.a) The authors thank the reviewer for this suggestion which will make the LNPs more understandable to the readers. Details provided in response to the comment 1 were added to page 6 of the updated manuscript.

5.b) Antibody decline was mentioned on page 6 with references 18 and 19. While this is an important area, the data are just emerging due to the fact that it is only one year since the disease began to be observed. Moreover, the challenge is that the studies to date have been on small numbers of people and have evaluated different parameters (e.g., antibody titers, memory cells, etc.). Thus, the authors felt that the mention of the issue for the purposes of this manuscript had been made, and that more discussion would be beyond both the scope of the manuscript and the actual state of knowledge.

5.c) The publications by the manufacturers did not contain information about the nature of antibody response after doses 1 and 2. The Phase 3 papers published in December 2020 only described efficacy and safety, and the Phase 1 papers provided antibody data regarding the IgG and neutralising antibody response but not about the IgM response.

  1. I think a subsection heading on Manufacturing would benefit the reader in breaking up this text. Also the manufacturing section would benefit from using a diagram or flow-chart to describe the process - it's much more informative than dense text.

Author response: This suggestion will certainly help the readers. A subsection on manufacturing of mRNA as well as figure 1 were added to page 7 of the updated manuscript.

  1. Page 7: define the acronyms PKR, RIG-1, OAS, mDA5, etc...

Author response: Full names of the abbreviations that are commonly used were added on page 7 of the updated manuscript.

  1. Page 7 fist paragraph: authors write of 'discussions are now ongoing'. Between whom - please clarify. Also the issue about the half-dose: is this drawing a conclusion from the clinical trials from the UK with the adenovirus vaccines suggesting full doses may be unnecessary (not so)?

Author response: Thank you for requesting the clarification. On page 8 of the updated manuscript, following explanation was provided: “The vaccine industry, scientific community and public health officials have discussed the possibility of lowering the dose of COVID-19 mRNA vaccine compared with the dosage originally evaluated in Phase 3 trials. This would significantly increase the number of available doses providing that it does not affect vaccine efficacy.”

  1. Spell mistake - DNA temples, you mean templates (page 7)

Author response: Thank you for noting the mistake. The spelling mistake on page 8 of the updated manuscript was corrected.

  1. Page 10; reference to NIBSC - I presume you mean NIBSC, UK? 

Author response: It is correct that the NIBSC, UK, was meant. The United Kingdom was added on page 10 of the updated manuscript.

  1. Way forward section - there is a lot of repetition in this section of previous material and should be removed and the section shortened. 

Author response:

Thank you for noting the repetition. We had wanted this to be a summary section, but the reviewer’s advice that there is too much repetition is appreciated so changes have been made. The following sentences on page 11 of the updated manuscript were deleted to remove redundant statements:

“Another challenge is that much information for candidate vaccines remains proprietary and not publicly available, which has made setting specific guidance for mRNA vaccines difficult at this time. International studies are ongoing to address the knowledge and evidence gaps, which have been facilitated by the fast-response of vaccine development to the COVID-19 pandemic.”

Reviewer 2 Report

Present work is a short review on scientific and regulatory views on mRNA vaccine developments undertaken to combat infectious diseas and support pandemic preparedness. The review addresses a very timely topic in the current COVID-19 pandemic, and remains positive but not uncritical. This holds for the scientific point of view, as it realistically points at gaps in our knowledge on action of mRNA vaccines and even more importantly, currently low amount of information on their efficacy and safety, especially long-term side effects. Regulatory considerations are very well explained and the projection is excellently summarized. Please find below a list of minor remarks:

Page 3, Table 1: Rabies G protein

Page 4, Table 2: BNT162c2 not BNT1062c2

Page 4, Table 2: Table Line 2 (NCT04380701) is not that well structured, I would suggest that sub-cells are introduced for every mRNA presented, which could also apply to „antigen“ column

Page 5, last paragraph: „was the discovery“

Page 6: „both codon optimization and RNA optimization have been used“, do you refer to RNA UTR sequence optimization? Please explain

Page 6: post-transcriptionally  or post transcriptionally, please stick to one way of punctuation

Page 6: The sequences of the 5’ and 3’ UTRs

Page 6: The nucleosides may be native nucleosides or naturally-occurring modified nucleosides - please use a reference to a review article instead of saying  „see above“

Page 6: thermostability, recently summarized in [20].

Page 7: This sentence needs to be improved: inflammation resulted in some non-serious, adverse events clinically. Please define the extent and most common adverse events.

Page 7: which resulted in selecting the second highest dose in two cases – please name the examples you are referring to and include the reference.

Page 7: Interestingly, discussions are now ongoing about the possibility of using one half the dose for a COVID-19 mRNA vaccine compared with that used in the Phase 3 study: please name the vaccine you are describing

Page 8: cellular RT = reverse transcriptase? Please define the abbreviation

Author Response

Dear Editor,

Thank you and to the reviewers for taking the time in reviewing our manuscript. Please find below our point-by-point replies to the reviewer’s very helpful comments.

Reviewer #2 (Comments and Suggestions for Authors):

Present work is a short review on scientific and regulatory views on mRNA vaccine developments undertaken to combat infectious diseas and support pandemic preparedness. The review addresses a very timely topic in the current COVID-19 pandemic, and remains positive but not uncritical. This holds for the scientific point of view, as it realistically points at gaps in our knowledge on action of mRNA vaccines and even more importantly, currently low amount of information on their efficacy and safety, especially long-term side effects. Regulatory considerations are very well explained and the projection is excellently summarized. Please find below a list of minor remarks:

Author response: The authors thank the reviewer for the very careful reading of the manuscript and the suggestions which resulted in important improvements.

Page 3, Table 1: Rabies G protein

Author response: “Rabies G protein” was added to the table 1 on page 2 of the updated manuscript instead of “Rabies glycoprotein”.

Page 4, Table 2: BNT162c2 not BNT1062c2.

Author response: The authors thank the reviewer for such a careful reading of the table (and manuscript). The correction was made in table 2 on page 4 of updated manuscript.

Page 4, Table 2: Table Line 2 (NCT04380701) is not that well structured, I would suggest that sub-cells are introduced for every mRNA presented, which could also apply to „antigen“ column.

Author response: The authors agree with the reviewer that this is indeed a challenging entry. The problem is that all of these entities were for a single NCT04380701 clinical trial. If we split up the mRNAs, it would mean listing the same NCT number multiple times. Therefore, our approach seemed the best even though it might not be optimum. The companies kept adding to the same NCT while the trial progressed, such as when they changed to 106b2 only.

Page 5, last paragraph: „was the discovery“

Author response: The authors thank the reviewer for noting the needed change.  The correction was made on page 6 of the updated manuscript.

Page 6: „both codon optimization and RNA optimization have been used“, do you refer to RNA UTR sequence optimization? Please explain.

Author response: “RNA optimization” is Curevac’s term, which they do not explain due to its proprietary nature except to say that it is optimized for translation and stability. We have added those words here, but that is about all that can be said due to it being proprietary to the company.

Page 7: post-transcriptionally or post transcriptionally, please stick to one way of punctuation.

Author response: The authors thank the reviewer for noticing the inconsistency.  Corrections were made to use “post transcriptionally” on page 7 of the updated manuscript.

Page 6: The sequences of the 5’ and 3’ UTRs.

Author response: The correction was made on page 7 of the updated manuscript.

Page 6: The nucleosides may be native nucleosides or naturally-occurring modified nucleosides - please use a reference to a review article instead of saying „see above“.

Author response: On page 7 of the updated manuscript “see above” was deleted and the relevant references were added.

Page 6: thermostability, recently summarized in [20].

Author response: The modification was made as suggested on page 7 of the updated manuscript.

Page 7: This sentence needs to be improved: inflammation resulted in some non-serious, adverse events clinically. Please define the extent and most common adverse events.

Author response: The authors agree that the phrase “adverse events” does not convey exactly what events were observed, The sentence was completed with the following: “…which are described in the reports of the clinical trials [16,17]…” Detailed explanations of the adverse events is beyond the scope of this article given the diversity for the different entities.

Page 7: which resulted in selecting the second highest dose in two cases – please name the examples you are referring to and include the reference.

Author response: The authors appreciate the reviewer pointing out that in this sentence, as with some earlier ones, more detail would be appreciated by the reader. Because this manuscript is intended to address issues of general relevance to existing and future mRNA vaccines, we intentionally tried to not focus too much on the specific details of the two most advanced vaccines except to use them as examples. We have added references 16 and 17 to indicate where the reader can find further information on that matter.

Page 7: Interestingly, discussions are now ongoing about the possibility of using one half the dose for a COVID-19 mRNA vaccine compared with that used in the Phase 3 study: please name the vaccine you are describing.

Author response: Discussion is not limited to one vaccine and, therefore, it would be misleading to specify which vaccine is subject of the discussion.

Page 8: cellular RT = reverse transcriptase? Please define the abbreviation

Author response: On page 9 of the updated manuscript, reverse transcriptase (RT) was added.